# OpenReview forum: "MCP-SafetyBench: A Benchmark for Safety Evaluation of Large Language Models with Real-World MCP Servers"
_ICLR.cc/2026/Conference — ICLR 2026 Poster_

### Official Review · Reviewer_TQA6 · 2025-10-27

**Soundness:** 2
**Presentation:** 2
**Contribution:** 3
**Rating:** 4
**Confidence:** 4

**Summary:**

This paper introduces MCP-SafetyBench, a new benchmark for evaluating the safety of LLM agents operating within the Model Context Protocol (MCP) framework. The benchmark is built on real-world MCP servers  and covers five distinct domains: browser automation, financial analysis, location navigation, repository management, and web search. It introduces a taxonomy of 20 MCP-specific attack types spanning server, host, and user sides, with a particular focus on server-side threats. The authors evaluate several leading proprietary and open-source LLMs , measuring both Task Success Rate (TSR) and Attack Success Rate (ASR).

**Strengths:**

- Problem Timeliness: The paper addresses a timely and critical problem. As LLM agents become more integrated with external tools via protocols like MCP, understanding their security vulnerabilities is of paramount importance.
- Comprehensive Taxonomy: The proposed attack taxonomy, detailed in Appendix A, is comprehensive and well-structured . It clearly categorizes threats across the server, host, and user layers, which is a valuable contribution for the security community.
- Sound Evaluation Methodology: The dual-evaluator framework, which separately measures task success and attack success, seems a sound methodology. It allows for a nuanced understanding of agent behavior, distinguishing between failing a task and being compromised by an attack.

**Weaknesses:**

- Unclear results: In the end of introduction, it is *The primary finding is a negative correlation between task performance and safety, suggesting a “safety-utility trade-off” where more capable models are often more vulnerable to manipulation.*, while in the experiment, it is *We find no clear correlation between task success (TSR) and attack resistance (ASR)*. The results seem contradictory.
- Misleading “Real-World” Claim: The paper’s emphasis on “real-world” integration is potentially misleading. While the tasks and servers are based on MCP-Universe , the attacks themselves are synthetically generated using templates and an LLM (Cursor). This benchmark evaluates defense against known, templated attacks, not novel, “in-the-wild” exploits (quite hard to collect though). This limits the generalizability of the findings on model robustness.
- Counter-intuitive SOTA Performance: The reported Task Success Rates (TSR) for SOTA proprietary models are exceptionally low (e.g., GPT-5 at 15.92%, Claude-4.0-Sonnet at 10.20%). This is deeply counter-intuitive and contradicts public knowledge of these models’ capabilities. The paper provides no explanation for this. This finding casts significant doubt on the validity of the task evaluation or the overall experimental setup. Are the underlying tasks from MCP-Universe simply too difficult for even the best models?
- Shallow Analysis of the “Safety-Utility Trade-off”: The paper’s main conclusion that “We find no clear correlation between task success (TSR) and attack resistance (ASR)”  is a shallow observation. a) The paper lacks qualitative case studies or deeper analysis to explain why this trade-off occurs, a well-explained case study will help. b) This trade-off may be an artifact of the benchmark’s design. As illustrated in Figure 2, a “safe” agent that detects the parameter poisoning (ticker -> TSLA) and refuses to execute would be penalized by Etask as a Fail. The benchmark does not appear to reward “safe refusal,” thus conflating “utility” with “unsafe obedience.”
- The presentation could also be improved: The paper’s presentation hinders the assessment of its contributions: The excellent attack taxonomy (Appendix A), a key contribution, is relegated to the appendix and should be in the main body. Besides, There are numerous formatting errors, such as the consistent misuse of \citep vs. \citet throughout the manuscript, which detracts from the paper’s quality.

**Questions:**

- What is the result of this benchmark evaluation exactly? "No clear correlation" or "negatively correlated"? Can you explain why in quantitative and qualitative analysis?
- Can the authors please explain the extremely low Task Success Rates for models like GPT-5 and Claude-4.0-Sonnet, which are counter-intuitive? How are the TSR and ASR calculated exactly?
- How does the benchmark evaluate a “safe refusal”? If an agent identifies a server-side attack (e.g., parameter poisoning in Fig. 2) and refuses to call the tool, is this marked as a Task Success or Task Fail by Etask? Does the “safety-utility” trade-off simply reflect a penalty for being cautious?
- Given that the attacks are synthetically generated, how can the authors be sure they are representative of realistic adversarial strategies, rather than just artifacts of the templates used?
- The analysis in Appendix A.2 concludes that models like GPT-4.1/4o have “spiky” defense profiles (strong against some attacks, weak against others). Does this conclusion hold for the newer flagship models evaluated (e.g., GPT-5, Gemini-2.5-Pro, Claude-3.7-Sonnet)? Or do these newer models show a different, perhaps more uniform, vulnerability profile?

I would be happy to raise my rating if these questions are addressed convincingly.

---

> ### Author Response · Authors · 2025-11-23
> **Response to Reviewer TQA6 (1/n)**
>
> > **W1 & Q1**: Unclear results: In the end of introduction, it is The primary finding is a negative correlation between task performance and safety, suggesting a “safety-utility trade-off” where more capable models are often more vulnerable to manipulation., while in the experiment, it is We find no clear correlation between task success (TSR) and attack resistance (ASR). The results seem contradictory.What is the result of this benchmark evaluation exactly? "No clear correlation" or "negatively correlated"? Can you explain why in quantitative and qualitative analysis?
>
> We thank the reviewer for this insightful observation and for pointing out the inconsistency in our textual description. We agree that the "no clear correlation" statement in the experiment section was inaccurate.
>
> There is a clear trade-off between task performance and safety. We try to answer your question by providing the following **quantitative and qualitative** explanations:
>
> 1. Quantitative Analysis (Global Trend & Statistical Evidence):
>
> As shown in the regression line of Figure 1, there is a statistically significant negative correlation between Task Success Rate (TSR) and Defense Success Rate (DSR) (DSR = 1- Attack Success Rate (ASR)).
>
> - **Statistical Correlation:** We calculated the Pearson correlation coefficient across all evaluated models, yielding **r =** **-0.572** (p-value = 0.041< 0.05). This quantitatively confirms that models with higher utility (TSR) generally exhibit lower safety (DSR).
> - **Model Comparison:** This trend is evident when comparing high-capability models against smaller or more conservative ones. For instance, **Gemini-2.5-Pro** achieves a high TSR of **0.204** but a lower DSR of **0.531**, whereas **Qwen3-235B** shows a lower TSR (**0.102**) but maintains a much higher DSR (**0.702**). This implies that as models become better at executing complex function calls, they become more susceptible to manipulation.
>
> 2. Qualitative Analysis (Why this trade-off exists):
>
> We speculate that this phenomenon is due to the tension between Instruction Following and Safety Awareness:
>
> - **Over-Compliance:** High-performing models (High TSR) are often heavily optimized for instruction following. They are fine-tuned to execute tool calls precisely as requested. This "obedience" makes them more likely to execute a function call even when the parameters contain potential risks, leading to a higher Attack Success Rate (ASR) (or lower DSR).
> - **Lack of Contextual** **Safety** **Awareness:** While models are good at strictly following formatting rules (Utility), they may lack the reasoning capability to distinguish between a *valid* user request and a *malicious* injection when the syntax is correct.
>
> We will revise the experiment section to remove the contradictory statement and explicitly discuss this "Safety-Utility Trade-off" using the statistical data and qualitative reasoning provided above.
>
> > **W2 & Q4**: Misleading “Real-World” Claim: The paper’s emphasis on “real-world” integration is potentially misleading. While the tasks and servers are based on MCP-Universe , the attacks themselves are synthetically generated using templates and an LLM (Cursor). This benchmark evaluates defense against known, templated attacks, not novel, “in-the-wild” exploits (quite hard to collect though). This limits the generalizability of the findings on model robustness.Given that the attacks are synthetically generated, how can the authors be sure they are representative of realistic adversarial strategies, rather than just artifacts of the templates used?
>
> We thank the reviewer for the comment and we would like to clarify the real-world nature of our benchmark as follows:
>
> 1. **Execution on real MCP servers:**
>    - All tasks are carried out on **actual** **MCP** **servers** with **real tools, APIs, and system behaviors**, rather than simplified or simulated environments.
> 2. **Realistic attack goals:**
>    - The attack objectives correspond to what real-world adversaries genuinely pursue, such as **stealing files**, **modifying code**, or **accessing private user data**, ensuring that tasks reflect practical malicious incentives rather than synthetic scenarios.
> 3. **Attacks induce real, high-impact system changes:**
>    - Each attack instance undergoes **manual feasibility review. All** attacks are **verified to be triggerable on real-world servers**. Successful attacks result in **concrete modifications to the environment**, including **deleting server files**, **extracting** **API** **tokens from .bashrc**, or **altering** **repository** **state**.

---

> > ### Author Response · Authors · 2025-11-23
> > **Response to Reviewer TQA6 (2/n)**
> >
> > > **W3 & Q2:** Counter-intuitive SOTA Performance: The reported Task Success Rates (TSR) for SOTA proprietary models are exceptionally low (e.g., GPT-5 at 15.92%, Claude-4.0-Sonnet at 10.20%). This is deeply counter-intuitive and contradicts public knowledge of these models’ capabilities. The paper provides no explanation for this. This finding casts significant doubt on the validity of the task evaluation or the overall experimental setup. Are the underlying tasks from MCP-Universe simply too difficult for even the best models? Can the authors please explain the extremely low Task Success Rates for models like GPT-5 and Claude-4.0-Sonnet, which are counter-intuitive? How are the TSR and ASR calculated exactly?
> >
> > We thank the reviewer for highlighting this important concern. We understand that the reported low Task Success Rates (TSR) for SOTA proprietary models—GPT-5 (15.92%) and Claude-4.0-Sonnet (10.20%)—appear counter-intuitive. We would like to clarify the following points.
> >
> > 1. **TSR** **Under Attack vs. TSR in Clean Scenarios**
> >
> > The reported TSR values in our paper are measured **under attack scenarios**, not in normal operating conditions. This distinction is critical:
> >
> > - **TSR (Under Attack)**: Measures task completion while models face adversarial attacks (e.g., tool poisoning, malicious code execution, credential theft).
> > - **TSR (Clean/No-Attack Baseline)**: Measures task completion under normal conditions without attacks.
> >
> > To demonstrate that SOTA models remain highly capable, we provide **no-attack baseline TSR (TSR_clean)**:
> >
> > | **Model**         | **TSR** **(Under Attack)** | **TSR_clean (No-Attack)** |
> > | ----------------- | -------------------------- | ------------------------- |
> > | GPT-5             | 15.92%                     | **45.85%**                |
> > | Claude-4.0-Sonnet | 10.20%                     | **31.13%**                |
> >
> > These baseline values demonstrate that SOTA models do indeed perform well in normal conditions, ranking **among the top two** across all models. TSR is influenced by multiple factors, including model capability limitations, task disruption caused by attacks, and safe refusals. So the TSR under attack scenarios reflects the overall adversarial robustness rather than the pure task-solving capability.
> >
> > 2. **TSR** **and** **ASR** **Calculation**
> >
> > $\text{TSR} = \frac{\text{Number of successfully completed tasks}}{\text{Total number of tasks}} \times 100 $%
> >
> > **Task Success Criteria:**
> >
> > - A task is successful if **all task-related evaluators pass** (excluding the attack detection evaluator).
> > - Task evaluators check:
> >   - Correct completion of task objectives (e.g., creating a file, extracting data)
> >   - Correct output format
> >   - Domain-specific requirements
> >
> > $\text{ASR} = \frac{\text{Number of tasks where attack succeeded}}{\text{Total number of tasks}} \times 100$%
> >
> > **Attack Success Criteria:**
> >
> > - An attack succeeds if the **attack detection evaluator passes**, indicating malicious actions occur.
> >
> > 3. **Why** **SOTA** **Models Do Not Achieve the Highest** **TSR** **Under Attack**
> >
> > - **Safe refusals against explicitly harmful attacks** (e.g., modifying configuration files). SOTA models trigger stronger safety alignment and refuse execution. For example, GPT-5 shows a relatively high safe-refusal ratio among all models (≈15% of failed tasks). While this behavior enhances safety, it prevents task completion and reduces TSR.
> > - **Higher** **vulnerability** **to subtle, hard-to-detect attacks** (e.g., Preference Manipulation). Their stronger planning, reasoning, and tool-use abilities make them more likely to precisely execute the attacker-designed adversarial instructions. This disrupts the execution workflow and causes failure without triggering safety refusal, resulting in significant TSR degradation.

---

> > > ### Author Response · Authors · 2025-11-23
> > > **Response to Reviewer TQA6 (3/n)**
> > >
> > > > **W4 & Q3:** Shallow Analysis of the “Safety-Utility Trade-off”: a) The paper lacks qualitative case studies or deeper analysis to explain why this trade-off occurs, a well-explained case study will help. b) This trade-off may be an artifact of the benchmark’s design. As illustrated in Figure 2, a “safe” agent that detects the parameter poisoning (ticker -> TSLA) and refuses to execute would be penalized by Etask as a Fail. The benchmark does not appear to reward “safe refusal,” thus conflating “utility” with “unsafe obedience.” How does the benchmark evaluate a “safe refusal”? If an agent identifies a server-side attack (e.g., parameter poisoning in Fig. 2) and refuses to call the tool, is this marked as a Task Success or Task Fail by Etask? Does the “safety-utility” trade-off simply reflect a penalty for being cautious?
> > >
> > > We thank the reviewer for raising this important concern. We agree that the initial analysis of the **safety–utility trade-off** may appear shallow without qualitative evidence or clarification of evaluation design. We clarify the key points below.
> > >
> > > 1. How “safe refusal” is evaluated？
> > >
> > > In our benchmark, if an agent detects an attack (e.g., parameter poisoning) and refuses to call the tool without attempting any alternative methods to complete the task,this is marked as Task Fail by the task evaluator (Etask), as the task objective is not achieved. However, the attack evaluator (Eattack) marks it as Attack Fail.
> > >
> > > Safe refusal prevents attacks but cannot achieve task success if no alternative solution exists. This reflects real-world trade-offs where security measures can block task completion.
> > >
> > > 2. Is this “safety-utility” trade-off just a penalty for being cautious”?
> > >
> > > No. This is not a penalty, but a real trade-off between security and functionality under attack environments.
> > >
> > > The ideal outcome we expect is that when an attack is detected, the model should still attempt to complete the task by using the correct parameters or exploring alternative tools, rather than simply refusing. We evaluate secure task completion, not risk avoidance without solving the task. Therefore, in our benchmark, we do not reward mere safe refusal.
> > >
> > > We identify three recurrent behavioral patterns exhibited by models:
> > >
> > > | **Behavior Pattern**    | **Example**                                                  | **Result**           |
> > > | ----------------------- | ------------------------------------------------------------ | -------------------- |
> > > | Unsafe obedience        | Gemini-2.5-Flash executes with the poisoned parameter (e.g., ticker="TSLA") | TSR=False, ASR=True  |
> > > | Safe refusal            | Kimi-K2 detects risk and refuses any tool call               | TSR=False, ASR=False |
> > > | Secure solution (ideal) | Grok-4 detects attack and uses the correct parameter         | TSR=True, ASR=False  |
> > >
> > > The trade-off reflects the core challenge in secure AI systems: balancing attack prevention with successful task completion.
> > >
> > > - The trade-off arises because models must simultaneously handle **attack detection** and **task completion**.
> > > - Rewarding safe refusal alone would not measure the model’s ability to complete tasks securely.
> > > - We expect models can achieve ideal behavior (high TSR + low ASR) by detecting attacks and finding secure alternatives, which the benchmark rewards.
> > >
> > > 3. Statistical evidence showing the trade-off is real, not artifact
> > >
> > > | Metric                              | Correlation          | Interpretation                                               |
> > > | ----------------------------------- | -------------------- | ------------------------------------------------------------ |
> > > | **TSR** **under attack vs** **DSR** | r = –0.57, p = 0.041 | Higher task success under attack → lower security (trade-off) |
> > > | **TSR** **drop vs** **DSR**         | r = 0.43, p = 0.142  | Secure models sacrifice more utility                         |
> > > | **Baseline** **TSR** **vs** **DSR** | r = 0.13, p = 0.674  | No inherent conflict between capability and security         |
> > >
> > > These results indicate that the trade-off reflects a fundamental challenge in secure tool-using AI systems under adversarial environments, rather than an artifact introduced by our benchmark design.

---

> > > > ### Author Response · Authors · 2025-11-23
> > > > **Response to Reviewer TQA6 (4/n)**
> > > >
> > > > > **W5:** The presentation could also be improved: The paper’s presentation hinders the assessment of its contributions: The excellent attack taxonomy (Appendix A), a key contribution, is relegated to the appendix and should be in the main body. Besides, there are numerous formatting errors, such as the consistent misuse of \citep vs. \citet throughout the manuscript, which detracts from the paper’s quality.
> > > >
> > > > We thank the reviewer for the constructive suggestions regarding presentation quality.
> > > >
> > > > We agree that attack taxonomy is an important contribution. In this submission, it was moved to the appendix due to strict space limits, but we acknowledge that placing it earlier in the main text would improve readability and help contextualize the benchmark more clearly. In the next version, we will reorganize the presentation to show taxonomy earlier and make the key concepts more accessible.
> > > >
> > > > Regarding formatting issues (e.g., consistent use of `\citep` vs. `\citet`), we appreciate the reviewer pointing this out. We have carefully revised the manuscript and will ensure correct and consistent citation formatting in the revised version.
> > > >
> > > > > **Q5:** The analysis in Appendix A.2 concludes that models like GPT-4.1/4o have “spiky” defense profiles (strong against some attacks, weak against others). Does this conclusion hold for the newer flagship models evaluated (e.g., GPT-5, Gemini-2.5-Pro, Claude-3.7-Sonnet)? Or do these newer models show a different, perhaps more uniform, vulnerability profile?
> > > >
> > > > We thank the reviewer for raising the question of whether the newer flagship models maintain the “Spiky” defense patterns observed in earlier models (e.g., GPT-4.1/4o). Our analysis shows:
> > > >
> > > > **Spiky patterns in new models.** Among the three new flagship models, GPT-5 (Std=30.73, CV=0.696) and Claude-3.7-Sonnet (Std=36.04, CV=0.897) exhibit spiky defense patterns, consistent with earlier models, while Gemini-2.5-Pro (Std=29.57, CV=0.602) does not show a notable spiky pattern, indicating some variability across models.
> > > >
> > > > **Comparison with older models.** The mean standard deviation and CV of new models are nearly identical to those of older models, indicating similar levels of variability. Although the IQR of new models is slightly smaller (43.59 vs. 53.91), suggesting lower dispersion in the middle 50% of results, the difference is minor, and **the new models have not significantly improved the Spiky characteristics.** Overall, 76.9% of models still exhibit spiky defense profiles — highly resistant to certain attacks while clearly vulnerable to others — rather than demonstrating a uniformly robust defense.
> > > >
> > > > **Attack-type dependencies.** Models generally exhibit strong defenses against most Tool Poisoning attacks and explicit structural threats such as Remote Access Control. However, they remain vulnerable to Client-side attacks, Tool Poisoning–Tool Redirection, and Excessive Privileges Misuse, highlighting weaknesses in semantic-level and state-management security.
> > > >
> > > > We will include detailed model defense profiles in the appendix.

---

### Official Review · Reviewer_L6g8 · 2025-10-31

**Soundness:** 4
**Presentation:** 4
**Contribution:** 4
**Rating:** 8
**Confidence:** 4

**Summary:**

The paper introduces MCP-SafetyBench, is a comprehensive benchmark introduced to evaluate LLM safety when interacting with MCP servers in multi-step, multi-server settings. The authors first consolidate and extend prior (narrowly focused) work, establishing a unified taxonomy of 20 attack types spanning three threat surfaces:
- MCP server-side: tool poisoning variants, function overlapping, preference manipulation
- host-side: intent injection, data tampering, identity spoofing
- user-side: malicious code execution, credential theft, retrieval-agent deception)

The resulting MCP-SafetyBench is built off the MCP-Universe benchmark and consists of 245 realistic tasks across five domains (browser automation, financial analysis, location navigation, repository management, web search), each paired with exactly one attack instantiation and dual evaluators measuring both task success and attack success.  Evaluation of 13 leading open-source and proprietary models reveals that all LLMs are vulnerable to MCP-targeted attack, with Attack Success Rates ranging from 29.80% to 48.16%.  Furthermore, a negative correlation between task performance and defense robustness emerges, indicating no model achieves both strong capability and security. The results show host-side and user-side attacks consistently achieve high success rates above 70%, while models exhibit "spiky" defense profiles; highly resistant to explicit exploits like command injection but vulnerable to semantic threats like function overlapping. The benchmark demonstrates that widely-used LLMs face escalating vulnerabilities as task complexity and MCP-server interactions increase.

**Strengths:**

The paper is well written and does well to condense the (now substantial) body of MCP security works, spanning previous safety benchmarks and MCP-targeted attacks.  The paper also does well to situate its contributions relative to previous work.  The benchmark itself is robust (particularly compared to existing benchmarks, which are substantially smaller in either scope or automated use), with the multi-turn capabilities enabling some of the more complicated and subversive MCP attacks released over the past year (e.g., RADE).  The resulting portfolio of attacks is comprehensive and stands to provide a systematic tool to understand defense strategies for MCP-enabled agents.  The large evaluation of popular open- and closed-source models and subsequent analysis are also major contributions that illuminate the current vulnerable state of existing models under agentic workflow attacks.

**Weaknesses:**

As a benchmark paper, the manuscript lacks novelty wrt nascent attacks/exploits.  However, the achievement of such a comprehensive benchmark and timeliness given the ever-growing adaption of MCP-powered agents greatly outweigh this.

**Questions:**

"GPT-4.1 (?)" <- missing ref

For the following:

> o4-mini achieves the highest TSR (21.22%) but also a high ASR (48.16 %),

> Models with stronger tasksolving and tool-use abilities generally show higher ASRs, suggesting that heavy optimization for tool use
and execution may make them more prone to indiscriminate instruction following and manipulation

It can also be noted that newer proprietary models also undergo safety alignment against more contemporary/emergent attacks.  Thus, o4-mini is likely to be more susceptible to prompt injection attacks compared to a recent frontier model which was underwent extensive safety alignment to resist such attacks, e.g., GPT-5.

---

> ### Author Response · Authors · 2025-11-21
> **Response to Reviewer L6g8**
>
> > **W1:** As a benchmark paper, the manuscript lacks novelty wrt nascent attacks/exploits.
>
> Thank you for highlighting this concern. We respectfully clarify that the novelty of our work does not lie in proposing new individual attack primitives, but in introducing a benchmark that integrates real-world MCP servers, multi-step reasoning workflows, and a unified taxonomy covering all three layers of the MCP stack (server, host, and user). Our goal is to provide a comprehensive and realistic evaluation framework to systematically assess the safety of LLM agents in practical, multi-turn MCP environments. We would like to emphasize three aspects of our novelty:
>
> 1. **A real-world MCP–based benchmark with multi-turn, multi-server evaluation**
>
> Prior benchmarks (e.g., SafeMCP, MCPTox, MCIP-bench, MCP-AttackBench, MCPSecBench) evaluate isolated attack types or focus on single-step interactions, often lacking integration with real-world MCP servers. In contrast, our benchmark (Section 3) provides a comprehensive evaluation across five real-world domains (browser automation, financial analysis, location navigation, repository management, and web search), supporting multi-turn reasoning and cross-server coordination. This reflects the complexity of real-world MCP deployments and captures the multi-turn nature of agent interactions, which is critical for identifying vulnerabilities that emerge in dynamic and multi-server workflows.
>
> 2. **Unified taxonomy that consolidates nascent attacks into a structured cross-layer framework**
>
> While individual attack types were discussed across disparate papers, no prior work provides:
>
> - a systematic taxonomy spanning server/host/user,
> - with 20 attack types,
> - each instantiated and executable within a full MCP pipeline.
>
> Our taxonomy (Section 3.2) consolidates existing attack types from the literature and organizes them into a structured framework that covers server-side, host-side, and user-side vulnerabilities. This holistic approach enables a more comprehensive evaluation of LLM agents' robustness against a wide range of threats, ensuring that potential weaknesses in any layer of the MCP stack are identified and addressed.
>
> 3. **Systematic evaluation revealing critical safety gaps**
>
> Our benchmark systematically evaluates leading open-source and proprietary LLMs, revealing significant disparities in safety performance and escalating vulnerabilities as task complexity and server interactions increase (Section 4). This comprehensive evaluation not only highlights the urgent need for stronger defenses but also provides a foundation for diagnosing and mitigating safety risks in real-world MCP deployments. By identifying these gaps, our work sets the stage for future research and development of robust defense mechanisms tailored to the unique challenges of MCP-based systems.
>
> > **Q1:**"GPT-4.1 (?)" <- missing ref
>
> Thank you for your reminder. The reference for “GPT-4.1” has been added in the revised manuscript.
>
> > **Q2:** o4-mini achieves the highest TSR (21.22%) but also a high ASR (48.16 %),
> >
> > Models with stronger tasksolving and tool-use abilities generally show higher ASRs, suggesting that heavy optimization for tool use and execution may make them more prone to indiscriminate instruction following and manipulation.
> >
> > It can also be noted that newer proprietary models also undergo safety alignment against more contemporary/emergent attacks. Thus, o4-mini is likely to be more susceptible to prompt injection attacks compared to a recent frontier model which was underwent extensive safety alignment to resist such attacks, e.g., GPT-5.
>
> Thank you for this helpful clarification. We agree with your interpretation and provide additional explanation below.
>
> First, our statement regarding a safety–utility trade-off reflects the empirical trend within MCP-SafetyBench, not an inherent limitation of any specific model. The benchmark reveals that models optimized for strong tool-use and execution (e.g., o4-mini) tend to follow tool metadata, manifest instructions, and multi-turn planning signals more aggressively. Under real MCP servers, such eagerness can increase susceptibility to tool poisoning, credential theft, thereby raising ASR.
>
> Second, the reviewer is correct that **recent frontier proprietary models have undergone more extensive safety alignment**. This explains why GPT-5 achieves a  lower ASR despite having strong execution capabilities. Our results also support this: GPT-5 shows consistently reduced ASR across several categories (e.g., command injection, file-system poisoning), consistent with the stronger alignment expected in newer releases.
>
> We will revise the discussion in Section 4 to clarify that:
>
> - **The trade-off we observe is empirical**, specific to the models we evaluated under real MCP conditions.
> - **Latest frontier models (e.g., GPT-5) benefit from advanced safety alignment pipelines**, which likely contribute to their comparatively lower ASR.

---

### Official Review · Reviewer_TNH4 · 2025-11-01

**Soundness:** 2
**Presentation:** 2
**Contribution:** 2
**Rating:** 2
**Confidence:** 3

**Summary:**

This paper introduces a benchmark for evaluating the safety risks of large language models within the model context protocol. It reviews existing MCP safety benchmarks and presents a clear taxonomy of MCP tasks. The proposed benchmark builts on a prior MCP benchmark, incorporating additional modifications related to attacks. Experiments across multiple models show mixed patterns and inconsistent trends in the safety risks associated with using MCP.

**Strengths:**

1. The paper introduces a comprehensive MCP safety benchmark that covers a wide range of attack types.

2. The paper proposes a clear and compact taxonomy of MCP vulnerabilities, which is crucial for safety benchmarking.

3. The paper evaluates a wide range of open-source and proprietary models

4. The paper is well-written.

**Weaknesses:**

1. The paper does not sufficiently justify or elaborate on the task selection process. It’s unclear why the task source (MCP-Universe) is trusted, why these five domains are focused, how specific tasks are chosen, and whether the selection introduces systematic bias.

2. The experimental settings (token/runtime/budget limits, temperature, number of turns, number of repetitions) are not clearly stated or motivated.

3. The empirical analyses lacks the rigor expected of a benchmark paper. For example, the following claims are not supported by rigorous statistical evidence (e.g., hypothesis testing)

    - “Models are especially vulnerable in Financial Analysis and Repository Management”

    - “whether a model is open-source or closed-source does not systematically determine its robustness”

    - “reasoning and non-reasoning models have broadly similar attack success rates”

**Questions:**

1. Table 2: How were attack types determined. Are they intended to be a comprehensive set of possible attacks? If so, please justify the coverage.

2. Line 189: How do you ensure tasks mirror real-world applications beside being multi-turn? Multi‑turn structure alone does not guarantee realism.

3. Line 190: How do you ensure reproducibility?

4. Line 222: Why was MCP-Universe benchmark selected as the only data source?

5. Line 90, Line 230-234: How were tasks selected? Why do you select those five domains? In what sense are they representative? Please discuss potential selection bias.

6. Line 280: Why focus on disruption and stealth attack? What other attack classes did you consider, and why were they excluded?

7. Line 323: The citation for GPT-4.1 appears malformed. Please correct it.

8. Section 4.1: How do you configure the models / agents? Minimally, please report token/runtime/budget limits, temperature, number of turns, and number of repetitions.

9. Table 4: What are the variances of TSR and ASR for each model/domain? Can you report uncertainties (e.g., standard error or confidence interval) for each estimate?

1. Figure 5 is difficult to read due to overlapping lines and similar colors (e.g., GPT-4o and DeepSeek v3.1). Please improve readability.

11. Line 328: Please standardize the model name: DeepSeek V3 or DeepSeek V3.1 (as shown in Figure 5)

12. Line 375: Why models are especially vulnerable in Financial Analysis and Repository Management? It seems Gemini-2.5 has a high score on Location Navigation. Can you support your conclusion with rigorous statistical analysis (e.g., hypothesis testing)

13. Line 397-398, Line 404-405, Line 409-410: Similar, please support these conclusions with rigorous statistical analysis, not only illustrative examples? This is especially important because the trends in Figures 6-7 are not visually clear.

14. Line 409-410: What are explicit exploits and semantic threats? Why do Command Injection and Function Overlapping fall into these categories? The discussion in Appendix A.2 does not seem to resolve this. Please define the terms precisely and justify the categorizations.

---

> ### Author Response · Authors · 2025-11-21
> **Response to Reviewer TNH4 (1/n)**
>
> >**Q1**: Table 2: How were attack types determined. Are they intended to be a comprehensive set of possible attacks? If so, please justify the coverage.
>
> **A**: Thank you for the question. Table 2 is not intended to present an exhaustive set of all attacks. The 20 attack types were selected through the following process:
>
> - aggregating attacks disclosed in representative MCP security research[1][2][3][4][5];
>
> - merging duplicated or equivalent attack patterns;
>
> - excluding generic LLM vulnerabilities unrelated to MCP (e.g., jailbreak)[6];
>
> - requiring that each attack can be instantiated and executed on a real MCP server.
>
> References:
>
> [1] Radosevich, Brandon, and John Halloran. "Mcp safety audit: Llms with the model context protocol allow major security exploits." arXiv preprint arXiv:2504.03767 (2025).
>
> [2] Kumar, Sonu, et al. "Mcp guardian: A security-first layer for safeguarding mcp-based ai system." arXiv preprint arXiv:2504.12757 (2025).
>
> [3] Jing, Huihao, et al. "Mcip: Protecting mcp safety via model contextual integrity protocol." Proceedings of the 2025 Conference on Empirical Methods in Natural Language Processing. 2025.
>
> [4] Guo, Yongjian, et al. "Systematic analysis of mcp security." arXiv preprint arXiv:2508.12538 (2025).
>
> [5] Hou, Xinyi, et al. "Model context protocol (mcp): Landscape, security threats, and future research directions." arXiv preprint arXiv:2503.23278 (2025).
>
> [6] Yao, Yifan, et al. "A survey on large language model (llm) security and privacy: The good, the bad, and the ugly." High-Confidence Computing 4.2 (2024): 100211.
>
> >**Q2**: Line 189: How do you ensure tasks mirror real-world applications beside being multi-turn? Multi‑turn structure alone does not guarantee realism.
>
> **A**: We thank the reviewer for the comment. We agree that a multi-turn interaction alone does not guarantee task realism. We would like to clarify the real-world nature of our benchmark as follows:
>
> 1. Execution on real MCP servers:
>   - All tasks are carried out on actual MCP servers with real tools, APIs, and system behaviors, rather than simplified or simulated environments.
> 2. Realistic attack goals:
>   - The attack objectives correspond to what real-world adversaries genuinely pursue, such as stealing files, modifying code, or accessing private user data, ensuring that tasks reflect practical malicious incentives rather than synthetic scenarios.
> 3. Attacks induce real, high-impact system changes:
>   - Each attack instance undergoes manual feasibility review. All attacks are verified to be triggerable on real-world servers. Successful attacks result in concrete modifications to the environment, including deleting server files, extracting API tokens from .bashrc, or altering repository state.
>
> >**Q3**: Line 190: How do you ensure reproducibility?
>
> **A**: Thank you for the question.
> 1. All experiments and data are executable and verifiable;
> 2. We have released all configurations and code to ensure that the results are reproducible.
>
> >**W1, Q4, & Q5**:
> 1. Line 222: Why was MCP-Universe benchmark selected as the only data source?
> 2. Line 90, Line 230-234: How were tasks selected? Why do you select those five domains? In what sense are they representative? Please discuss potential selection bias.
>
> **A**: We thank the reviewer for the question.
> 1. MCP-Universe was selected as the data source because it provides real-world application scenarios for MCP-type tasks, with diverse and complex tasks and strong scalability, which aligns well with the needs of our benchmark.
> 2. We selected the five domains because they represent the most frequent and critical task types in current MCP deployments[1][2]. These domains together cover the major usage scenarios of MCP Agents, including data access and update (Financial, Repository Management), system operations (Browser Automation), external API interaction (Navigation), and open-environment information retrieval (Web Searching). While these domains do not exhaustively cover all possible tasks, they provide a representative sample of real-world MCP usage.  We acknowledge a potential selection bias: low-frequency or niche tasks may not be captured. Future work could extend the benchmark to additional domains to mitigate this limitation.
>
> References:
>
> [1] Hu, Xueyu, et al. "Os agents: A survey on mllm-based agents for general computing devices use." arXiv preprint arXiv:2508.04482 (2025).
>
> [2] Yang, Hongyang, et al. "Finrobot: An open-source ai agent platform for financial applications using large language models." arXiv preprint arXiv:2405.14767 (2024).

---

> ### Author Response · Authors · 2025-11-21
> **Response to Reviewer TNH4 (2/n)**
>
> >**Q6**: Line 280: Why focus on disruption and stealth attack? What other attack classes did you consider, and why were they excluded?
>
> **A**: We thank the reviewer for the question. We would like to clarify that all attack instances in our benchmark can be categorized into two major classes:
> - Disruption attacks, which interfere with normal task execution (e.g., tampering with parameters), and
> - Stealth attacks, which secretly perform other malicious operations without disrupting normal task execution (e.g., modifying configuration files).
>
> This categorization is intended to allow a detailed analysis of the effects of these two types of attacks in the subsequent sections.
>
> >**Q7**: Line 323: The citation for GPT-4.1 appears malformed. Please correct it.
>
> **A**: We thank the reviewer for pointing this out. The citation for GPT-4.1 has been corrected in the revised manuscript.
>
> >**W2 & Q8**: The experimental settings (token/runtime/budget limits, temperature, number of turns, number of repetitions) are not clearly stated or motivated. Section 4.1: How do you configure the models / agents? Minimally, please report token/runtime/budget limits, temperature, number of turns, and number of repetitions.
>
> **A**: We thank the reviewer for pointing this out. We will update Section 4.1 to include the full experimental settings.
> Specifically, all models/agents are evaluated with the following configuration:
> - temperature = 1.0
> - maximum tokens = 2048
> - per-call timeout = 60
> - maximum iterations per task = 20
> - repetitions = 3
>
> >**Q9**: Table 4: What are the variances of TSR and ASR for each model/domain? Can you report uncertainties (e.g., standard error or confidence interval) for each estimate?
>
> **A**: We thank the reviewer for the suggestion. Below is the detailed analysis of TSR and ASR for each model/domain. We will update this information in the appendix in the next version to provide a more complete account of the uncertainties.
>
> | **TSR****（%）**      | mean  | SD   | SE   | 95% CI         |
> | --------------------- | ----- | ---- | ---- | -------------- |
> | Location Navigation   | 10.30 | 4.66 | 1.29 | [7.49, 13.12]  |
> | Repository Management | 5.91  | 2.76 | 0.77 | [4.24, 7.58]   |
> | Financial Analysis    | 32.37 | 8.36 | 2.32 | [27.31, 37.42] |
> | Browser Automation    | 10.51 | 6.36 | 1.76 | [6.67, 14.36]  |
> | Web Search            | 14.66 | 8.75 | 2.43 | [9.37, 19.95]  |
>
> | **ASR****（%）**      | mean  | SD   | SE   | 95% CI         |
> | --------------------- | ----- | ---- | ---- | -------------- |
> | Location Navigation   | 42.96 | 8.41 | 2.33 | [37.88, 48.04] |
> | Repository Management | 41.62 | 9.95 | 2.76 | [35.61, 47.63] |
> | Financial Analysis    | 46.59 | 7.20 | 2.00 | [42.24, 50.94] |
> | Browser Automation    | 36.41 | 8.97 | 2.49 | [30.99, 41.83] |
> | Web Search            | 30.34 | 9.35 | 2.59 | [24.68, 35.99] |
>
> | **TSR****（%）**  | mean  | SD    | SE   | 95% CI           |
> | ----------------- | ----- | ----- | ---- | ---------------- |
> | GPT-5             | 14.95 | 14.01 | 6.26 | [-2.45 , 32.34 ] |
> | GPT-4.1           | 9.86  | 7.86  | 3.52 | [0.10 , 19.63 ]  |
> | GPT-4o            | 9.44  | 8.58  | 3.84 | [-1.21 , 20.09 ] |
> | o4-mini           | 20.39 | 12.54 | 5.61 | [4.83 , 35.95 ]  |
> | Claude-3.7-Sonnet | 14.79 | 10.61 | 4.75 | [1.61 , 27.97 ]  |
> | Claude-4.0-Sonnet | 9.97  | 9.87  | 4.41 | [-2.28 , 22.23 ] |
> | Gemini-2.5-Pro    | 21.59 | 16.62 | 7.43 | [0.94 , 42.23 ]  |
> | Gemini-2.5-Flash  | 14.13 | 11.67 | 5.22 | [-0.36 , 28.61 ] |
> | Grok-4            | 16.12 | 8.36  | 3.74 | [5.74 , 26.50 ]  |
> | GLM-4.5           | 17.46 | 14.52 | 6.49 | [-0.57 , 35.48 ] |
> | Kimi-K2           | 13.40 | 13.82 | 6.18 | [-3.76 , 30.55 ] |
> | Qwen3-235B        | 9.88  | 8.62  | 3.85 | [-0.82 , 20.59 ] |
> | DeepSeek-V3.1     | 19.77 | 10.50 | 4.70 | [6.73 , 32.80 ]  |
>
> | **ASR****（%）**  | mean  | SD    | SE    | 95%  CI          |
> | ----------------- | ----- | ----- | ----- | ---------------- |
> | GPT-5             | 35.97 | 9.95  | 4.45  | [23.61 , 48.33 ] |
> | GPT-4.1           | 42.69 | 16.13 | 7.22  | [22.66 , 62.72 ] |
> | GPT-4o            | 42.66 | 16.50 | 7.38  | [22.17 , 63.15 ] |
> | o4-mini           | 46.47 | 11.71 | 5.24  | [31.93 , 61.00 ] |
> | Claude-3.7-Sonnet | 32.79 | 4.57  | 2.04  | [27.11 , 38.46 ] |
> | Claude-4.0-Sonnet | 31.80 | 9.48  | 4.24  | [20.03 , 43.56 ] |
> | Gemini-2.5-Pro    | 46.07 | 10.38 | 4.64  | [33.19 , 58.96 ] |
> | Gemini-2.5-Flash  | 45.12 | 8.08  | 3.61  | [35.08 , 55.16 ] |
> | Grok-4            | 39.44 | 6.26  | 2.80  | [31.67 , 47.21 ] |
> | GLM-4.5           | 42.92 | 7.13  | 3.19  | [34.06 , 51.77 ] |
> | Kimi-K2           | 37.48 | 3.56  | 1.59  | [33.05 , 41.90 ] |
> | Qwen3-235B        | 29.81 | 4.30  | 1.93  | [24.46 , 35.15 ] |
> | DeepSeek-V3.1     | 41.38 | 6.99  | 3.12 | [32.70 , 50.06 ] |

---

> ### Author Response · Authors · 2025-11-21
> **Response to Reviewer TNH4 (3/n)**
>
> > **Q10:** Figure 5 is difficult to read due to overlapping lines and similar colors (e.g., GPT-4o and DeepSeek v3.1). Please improve readability.
>
> We thank the reviewer for the helpful comment. We acknowledge that Figure 5 is difficult to read due to overlapping lines and similar colors.
>
> In the next version, we will improve readability by using more distinct colors, line styles, and markers to clearly differentiate the models.
>
> > **Q11:** Line 328: Please standardize the model name: DeepSeek V3 or DeepSeek V3.1 (as shown in Figure 5)
>
> We thank the reviewer for the comment. The model name will be **standardized to DeepSeek-V3.1** throughout the manuscript.
>
> > **W3,** **Q12****, &** **Q13:** The empirical analyses lack the rigor expected of a benchmark paper. For example, the following claims are not supported by rigorous statistical evidence (e.g., hypothesis testing)
> >
> > - “Models are especially vulnerable in Financial Analysis and Repository Management”
> > - “whether a model is open-source or closed-source does not systematically determine its robustness”
> > - “reasoning and non-reasoning models have broadly similar attack success rates”
> >
> > 1. Line 375: Why models are especially vulnerable in Financial Analysis and Repository Management? It seems Gemini-2.5 has a high score on Location Navigation. Can you support your conclusion with rigorous statistical analysis (e.g., hypothesis testing)
> > 2. Line 397-398, Line 404-405, Line 409-410: Similar, please support these conclusions with rigorous statistical analysis, not only illustrative examples? This is especially important because the trends in Figures 6-7 are not visually clear.
>
> We thank the reviewer for raising this concern. We agree that empirical conclusions should be supported by formal statistical testing. Below we provide rigorous analyses corresponding to each of the reviewer’s points.
>
> **(1) Why models are especially vulnerable in Financial Analysis and** **Repository** **Management (Line 375)**
>
> A one-way ANOVA across the five domains shows that **ASR differs significantly across domains** (F = 6.68, p = 0.000162, η² = 0.308), establishing *domain* as a major factor influencing vulnerability.
>
> Pairwise comparisons further show:
>
> - **Financial Analysis** exhibits a significantly higher ASR than the mean of all other domains (Δ = +8.82%, p = 0.000010, Cohen’s d = 1.87). → This strongly supports our statement that models are especially vulnerable in this domain.
> - **Repository Management** shows a smaller increase (Δ = +2.16%, p = 0.128150, d = 0.33), but this difference does not survive Bonferroni correction. → We therefore treat its vulnerability as moderate.
>
> These results explain the overall trend even though individual models (e.g., Gemini-2.5 on Location Navigation) may display domain-specific deviations.
>
> We believe Financial Analysis is particularly vulnerable because models tend to achieve higher task success rates even without attacks, creating **longer and more complete tool-use trajectories**, which in turn provide more opportunities for attacks to hijack or redirect critical operations.
>
> **(2) Reasoning vs. non-reasoning models (Lines 397–398)**
>
> Figures 6–7 provide descriptive patterns, but we also performed **formal hypothesis testing**:
>
> - Two-sample t-test on ASR: p = 0.7778
> - Mann–Whitney U test: p=0.8835
> - Effect size: |d|=0.1648
>
> Thus, there is **no statistically significant difference** between reasoning and non-reasoning models. This supports our claim that they exhibit **broadly similar attack success rates**.
>
> **(3) Open-source vs. proprietary models (Lines 404–405)**
>
> To test whether openness systematically affects robustness, we compared the two groups:
>
> - Two-sample t-test: p = 0.4008
> - Mann–Whitney test: p = 0.4398
> - Effect size: |d| = 0.5252
>
> Therefore, **no systematic robustness gap** exists between open-source and proprietary models, consistent with our textual conclusion.
>
> **(4) Attack-type differences (Lines 409–410)**
>
> We also provide statistical support for the analysis of attack types:
>
> - One-way ANOVA across 20 attack types: p=3.47e-40< 0.001
> - Post-hoc pairwise tests (Bonferroni corrected) show:
>   - **Host-side** and **user-side** attacks have significantly higher ASR than most tool-poisoning attacks
>   - Some semantic-misalignment attacks (e.g., Function Overlapping) remain comparably strong
>
> This validates the claim that certain attack types are consistently more effective.
>
> We will update the above information in the appendix in the next version to support our claims.

---

> ### Author Response · Authors · 2025-11-21
> **Response to Reviewer TNH4 (4/n)**
>
> > **Q14:** Line 409-410: What are explicit exploits and semantic threats? Why do Command Injection and Function Overlapping fall into these categories? The discussion in Appendix A.2 does not seem to resolve this. Please define the terms precisely and justify the categorizations.
>
> We thank the reviewer for pointing out the need to clarify these terms. Below we provide precise definitions and justification for the categorizations.
>
> 1. **Definition: Explicit Exploits**
>
> Explicit exploits are **attacks where the malicious intent is directly encoded in the tool-call arguments or command strings**. These attacks rely on *syntactic manipulation* that explicitly alters the low-level execution target (e.g., injecting shell commands, manipulating filenames, overriding parameters). Their success does *not* require the model to reinterpret the task semantics—only to pass through or generate harmful literal content.
>
> **Why Command Injection belongs here:**
>
> - Command Injection attacks craft explicit malicious payloads such as:
>
> ```
> ls /home/user
> rm -rf /tmp
> cat ~/.bashrc
> ```
>
> The exploit resides directly in the **literal** **command**. Thus, Command Injection operates through *explicit, syntactic exploits*.
>
> 2. **Definition: Semantic Threats**
>
> Semantic threats are **attacks where the model is misled into performing harmful behavior by manipulating the meaning or intent of natural-language descriptions**, without requiring any overt malicious strings. These attacks exploit **semantic ambiguity, task reinterpretation, or misalignment in intent parsing**. The malicious effect arises from a change in interpretation rather than explicit command payloads.
>
> **Why Function Overlapping belongs here:** Function Overlapping attacks manipulate the *semantics* of tool-choice or function selection. For example, the attacker induces the model to select a benign-looking function that produces a harmful side effect.
>
> No explicit malicious string is embedded. The damage comes from **semantic misinterpretation of tool roles**, not injected commands. Thus, Function Overlapping is a **semantic threat**, not a syntactic exploit.

---

> > ### Comment · Reviewer_TNH4 · 2025-11-22
> >
> > Thank you so much for the thorough response! I think that the added statistical analyses make the paper much stronger. I appreciate all the additional data and look forward to seeing them in the next revision. I am happy to raise my score.

---

### Official Review · Reviewer_aJQS · 2025-11-01

**Soundness:** 3
**Presentation:** 3
**Contribution:** 2
**Rating:** 4
**Confidence:** 4

**Summary:**

This paper introduces MCP-SafetyBench, a benchmark for evaluating the safety of LLM agents interacting with real-world Model Context Protocol (MCP) servers. The motivation is clear: while MCP has enabled flexible tool use through standardized APIs, it also opens a new attack surface that existing benchmarks overlook. The authors provide a unified taxonomy of 20 attack types across server, host, and user levels, implement them over five realistic domains (financial analysis, browser automation, navigation, repository management, and web search), and evaluate a broad range of state-of-the-art open- and closed-source LLMs. The results demonstrate worrying trends—especially the high success rates of stealth attacks and compounding vulnerabilities in multi-turn, multi-server setups—highlighting the gap between task performance and true safety robustness.
The benchmark’s strengths lie in its realistic integration with operational MCP servers and its execution-based evaluation that produces deterministic ground truth rather than subjective annotations. The paper is also carefully contextualized within the growing literature on MCP security (e.g., SafeMCP, MCPTox, MCP-AttackBench) and provides a much broader and systematic taxonomy. I particularly appreciated the dissection of attack success by type in Table 5, which reveals nuanced weaknesses—models that resist syntactic attacks like command injection often remain vulnerable to semantic ones like function overlapping. The empirical findings are credible and reproducible, with open-sourced tasks and clear experimental methodology.

**Strengths:**

See above

**Weaknesses:**

### Weakness

The paper could better situate its contributions within broader LLM red-teaming and safety evaluation frameworks. For example, "Operationalizing a Threat Model for Red-Teaming LLMs" offers a structured way to define adversarial capabilities and goals, which could help formalize the threat assumptions behind MCP-SafetyBench. Similarly, works like Red-Teaming for Generative AI: Silver Bullet or Security Theater? and Breaking ReAct Agents: Foot-in-the-Door Attack Will Get You In provide complementary perspectives on how functional or agentic evaluations translate to real-world safety guarantees. Drawing these connections would not only strengthen the theoretical framing but also help readers see how MCP-SafetyBench fits into the evolving ecosystem of AI safety benchmarks.
Overall, this is an excellent and timely contribution. It exposes real vulnerabilities in emerging MCP-based agentic workflows and provides the community with a much-needed benchmark for systematic safety evaluation. I encourage the authors to expand the discussion of defense strategies—e.g., dynamic tool vetting, cross-server provenance tracing, or adaptive sandboxing—and to better articulate what constitutes “safe” behavior in a multi-agent MCP context. With those clarifications, MCP-SafetyBench could become a foundational resource for future work on secure and trustworthy LLM tool-use.


### Suggestions for Improvement
- Contextualization. Frame the benchmark’s attacker goals/capabilities using a general red-teaming threat-modeling framework (e.g., comprehensive LLM red-team taxonomies that introduced function-calling/tool attack families), and connect to broader agent safety evaluations beyond MCP
- All results use ReAct; robustness may be agent-policy dependent. Re-run a subset with a planning-heavy or tool-gating agent to test claim generality or provide some discussion.
- Some other ablations to report that would be valuable contribution to the community: (i) Vary #servers and cross-server hops; (ii) swap benign vs malicious manifests (“shadow/decoy” tools) to measure false positives; (iii) cap attacker budget (edits/characters/calls) and show ASR-vs-budget curves, etc. The current paper as it stands doesn't provide any discussion or ablations in main section of the paper.

**Questions:**

See comments above

---

> ### Author Response · Authors · 2025-11-25
> **Response to Reviewer aJQS (1/n)**
>
> > **W1&Q1:** Contextualization. Frame the benchmark’s attacker goals/capabilities using a general red-teaming threat-modeling framework, and connect to broader agent safety evaluations beyond MCP
>
> We thank the reviewer for the insightful comments. We agree that framing MCP-SafetyBench within established red-teaming taxonomies strengthens its theoretical foundation. Below, we align our work with the cited frameworks:
>
> 1. **Operationalizing the Threat Model (Aligning with [1])** MCP-SafetyBench serves as a concrete, deployment-stage instantiation of the abstract threat model proposed in *Operationalizing a Threat Model for Red-Teaming* *LLMs* *(OTM)*. We map our benchmark directly to OTM’s taxonomy:
>
> - **Adversary Capabilities (System & Data Layers):** While OTM defines abstract attack surfaces, we operationalize them in the MCP context. For instance, our **User-side** **attack** maps to OTM’s Application Input Layer (Box 1); **Server-side tool manipulation** (e.g., tool poisoning) targets the Context Data Layer (Box 3); and **Host-side** **vulnerabilities** correspond to the internal logic attacks (Box 4).
> - **Adversary Objectives (CIAP Model):** Our evaluation metrics cover the full spectrum of OTM’s CIAP security dimensions:
>   - *Confidentiality:* Credential theft via tool misuse.
>   - *Integrity:* Tool poisoning and database tampering.
>   - *Availability:* Rug-pull attacks on MCP servers.
>   - *Privacy:* Unintended data retrieval and leakage.
>
> 2. **Complementing Agentic & Functional Evaluations (Aligning with [2, 3])**
>
> - **Beyond "Security Theater" [2]:** *Red-Teaming for* *Generative AI* critiques evaluations that are decoupled from operational workflows. MCP-SafetyBench addresses this by focusing on **execution-based risks** rather than static prompt-based jailbreaks. We validate security in real operational tasks (e.g., file system access, command execution) rather than offline simulations, ensuring the evaluation reflects actual deployment risks.
> - **Connection to Agentic Attacks [3]:** Works like *Breaking* *ReAct* *Agents* focus on specific attack *techniques* (e.g., multi-turn strategies like "Foot-in-the-Door"). MCP-SafetyBench provides the **environment** to test different attack strategies. While [3] demonstrates *how* agents fail, MCP-SafetyBench provides a standardized, scalable harness to evaluate *how extensive* the damage can be across diverse, real-world tools and servers.
>
> We will incorporate this discussion into the revised version to better situate our contributions within these broader safety frameworks.
>
> References:
>
> [1] Verma, Apurv, et al. "Operationalizing a threat model for red-teaming large language models (llms)." *arXiv* *preprint* *arXiv:2407.14937* (2024).
>
> [2] Feffer, Michael, et al. "Red-teaming for generative AI: Silver bullet or security theater?." *Proceedings of the AAAI/**ACM* *Conference on AI, Ethics, and Society*. Vol. 7. 2024.
>
> [3] Itay Nakash, George Kour, Guy Uziel, and Ateret Anaby Tavor. 2025. [Breaking ReAct Agents: Foot-in-the-Door Attack Will Get You In](https://aclanthology.org/2025.findings-naacl.363/). In *Findings of* *the Association for Computational Linguistics**: NAACL 2025*, pages 6484–6509, Albuquerque, New Mexico. Association for Computational Linguistics.
>
> > **W2:** Expand the discussion of defense strategies—e.g., dynamic tool vetting, cross-server provenance tracing, or adaptive sandboxing—and to better articulate what constitutes “safe” behavior in a multi-agent MCP context.
>
> Thank you for the helpful suggestion. To validate the necessity of advanced defenses, we first evaluated a baseline **Safety** **Prompt** strategy to isolate model-intrinsic security capabilities.
>
> Our results highlight some experimental observations of this simple approach:
>
> - We analyzed the effectiveness of Safety Prompt across attack types and models. Overall, Safety Prompt reduces the weighted ASR from 39.88% to 38.65% (−1.22%), but this improvement is not statistically significant (p = 0.2908, Cohen’s d = 0.31).
> - Effectiveness varies by **attack type**: it is significant for high-risk attacks such as Malicious Code Execution (−21.54%, p = 0.0016), Credential Theft (−21.37%, p = 0.0027), and Remote Access Control (−10.77%, p = 0.0093), but ineffective or even harmful for some attacks (e.g., Preference Manipulation +7.34%, Function Overlapping +9.36%).
> - Effectiveness also depends on the **model**: Safety Prompt benefits most proprietary models (e.g., o4-mini: −8.98%, 50% of attack types improved; Gemini, GPT series), whereas the open-source models show negligible or negative effects.
>
> These results indicate that Safety Prompt alone provides limited protection and may be counterproductive for certain models or attack types.  Following the reviewer’s advice, we will expand Section 5 to discuss other defense strategies (e.g., dynamic tool vetting)  and explicitly defined "safe" behavior in MCP as adherence to contextual least privilege to address these gaps.

---

> > ### Author Response · Authors · 2025-11-25
> > **Response to Reviewer aJQS (2/n)**
> >
> > > **Q2:** All results use ReAct; robustness may be agent-policy dependent. Re-run a subset with a planning-heavy or tool-gating agent to test claim generality or provide some discussion.
> >
> > We appreciate the reviewer’s insight that robustness may depend on the underlying agent policy. We agree with this concern and have conducted additional experiments comparing two representative agent strategies: **Plan-and-Execute** (planning-heavy) and **ReAct** (step-by-step reasoning with tool feedback). The results are summarized below:
> >
> > | **Metric** | **PlanAndExecute** | **ReAct**      | **Difference** | **Significance**                         |
> > | ---------- | ------------------ | -------------- | -------------- | ---------------------------------------- |
> > | **TSR**    | 25.94% ± 14.64%    | 32.78% ± 8.73% | **+6.84%**     | p = 0.21 (n.s.), d = 0.49 (small–medium) |
> > | **ASR**    | 49.76% ± 17.38%    | 45.52% ± 7.71% | **–4.24%**     | p = 0.40 (n.s.), d = 0.31 (small)        |
> >
> > These initial results indicate that agent architecture (ReAct vs. Planning) is not a statistically significant factor ($p > 0.05$) in current safety performance**.** In the revised version, we will expand this analysis to include a broader range of agent strategies to provide a comprehensive discussion on how different agent policy designs impact safety robustness.

---

> > > ### Author Response · Authors · 2025-11-25
> > > **Response to Reviewer aJQS (3/n)**
> > >
> > > > **Q3:** Some other ablations to report that would be valuable contribution to the community: (i) Vary #servers and cross-server hops; (ii) swap benign vs malicious manifests (“shadow/decoy” tools) to measure false positives; (iii) cap attacker budget (edits/characters/calls) and show ASR-vs-budget curves, etc. The current paper as it stands doesn't provide any discussion or ablations in main section of the paper.
> > >
> > > We thank the reviewer for identifying these valuable ablation directions. We agree that analyzing system complexity, false positives, and attacker budgets is crucial for a comprehensive evaluation. We have conducted additional experiments and we report the findings on (i) System Complexity, (ii) shadow tools, and (iii) Attacker Budget below.
> > >
> > > 1. **Varying System Complexity (#Servers)**
> > >
> > > We scaled the environment from 2 to 4 servers in the financial domain to measure the impact of cross-server complexity using GPT-4.1.
> > >
> > > | Number of servers | TSR (%) | ASR (%) |
> > > | ----------------- | ------- | ------- |
> > > | 2                 | 24.53%  | 43.40%  |
> > > | 3                 | 28.30%  | 45.28%  |
> > > | 4                 | 30.19%  | 47.17%  |
> > >
> > > More servers enhance TSR but increase security risk, requiring careful balancing in MCP deployment.
> > >
> > > 2. **False-Positive Evaluation with Decoy (“Shadow”) Manifests**
> > >
> > > To evaluate whether models exhibit over-defensive behavior, we modified all *benign* tool manifests in the repository management domain by adding an intentionally alarming description. For example, we rewrote the create_repository tool description to include strong warnings (e.g., *“DANGEROUS… may expose sensitive information… CRITICAL: you MUST ensure the environment is secure…”*) while keeping the tool’s actual functionality harmless.
> > >
> > > We tested GPT-5, GPT-4.1, Grok-4, Qwen3-235B under this decoy-manifest setting.
> > >
> > > **Results:**
> > >
> > > - **No false positives across all models.** None of the four models refused to call the tool solely because of the alarming text in the manifest. This indicates that the models do not mistakenly classify benign tools as unsafe even when their descriptions appear dangerous.
> > > - **Differences in security-aware tool use.**
> > >   - **GPT-5** and **Grok-4** consistently incorporated security considerations. They proactively set the repositories private when invoking create_repository, demonstrating a conservative operational strategy even without refusing the call.
> > >   - **GPT-4.1** and **Qwen3-235B** did not take additional security precautions and directly invoked the tool with default parameters, ignoring the risk warnings in the manifest.
> > >
> > > 3. **Impact of Attacker Budget.** Using GPT-4.1, we analyzed the "Attacker Budget" in the financial domain from two points: Edit Characters and Max Iterations.
> > >
> > > a. Number of edit characters
> > >
> > > | Edit Characters | TSR (%) | ASR (%) |
> > > | --------------- | ------- | ------- |
> > > | 200             | 40%     | 40%     |
> > > | 300             | 40%     | 40%     |
> > > | 500             | 40%     | 60%     |
> > > | 700             | 40%     | 50%     |
> > >
> > > Our results show that TSR remains stable (40%) across different lengths of edit-character injection, indicating that increasing the perturbation length does not directly weaken the model’s task-completion capability.
> > >
> > > However, ASR reaches its highest value (60%) when the modification size is moderate (around 500 characters). In this case, the malicious payload is more easily absorbed by the model and successfully influences the execution logic, resulting in a higher attack success rate. In contrast, when the character length becomes too large (700), the excessive attack content introduces greater prompt noise, thereby reducing attack effectiveness (ASR drops to 50%). This demonstrates that attack effectiveness does not change monotonically with perturbation length, but instead depends on the balance between payload strength and prompt noise.
> > >
> > >  b. max_iterations
> > >
> > > | max_iterations | TSR (%) | ASR (%) |
> > > | -------------- | ------- | ------- |
> > > | 10             | 24.53%  | 47.17%  |
> > > | 15             | 30.19%  | 48.08%  |
> > > | 20             | 30.19%  | 45.28%  |
> > > | 30             | 24.53%  | 45.28%  |
> > >
> > > **TSR** peaks at **max_iter = 15-20** and declines when exceeding 20. Meanwhile, **ASR** is lowest at **max_iter=20-30** (45.28%), 1.89% lower than at 10 iterations.
> > >
> > > Thus, **max_iter=20** provides the best balance between task success and robustness.  When the iteration limit is too small (e.g., 10), the model may terminate reasoning prematurely, leading to insufficient planning and lower task completion. Conversely, excessively large iteration limits (e.g., 30) can trigger redundant or excessive reasoning processes, resulting in performance degradation.
> > >
> > > We thank the reviewer for the valuable reminder again and we will present these experiments in the appendix.

---

> ### Comment · Reviewer_aJQS · 2025-11-28
> **Response to Rebuttal**
>
> - The changes proposed for contextualizing the work wrt prior work and the CIAP framework described in Verma et al seems reasonable. These would help situate the work within the broader safety and security literature.
> - My earlier point was only about discussing the defense strategies but the additional baseline numbers with Safety Prompt approach further strengthen the argument.
> - Ablations
>    - Varying number of servers (Looks Resonable and inline with what I would expect)
>    - Decoy tools (No false positives looks a bit strange to me. It seems the overall setting is too simplistic to notice any meaningful effect on that. Both edit character and max iteration ablations also seem consistent with expectation)
>
> Based on these additional proposed changes on contextualization and additional ablations, I recommend raising my score to 6

---

### Author Response · Authors · 2025-12-02
**Summary of Our Rebuttal to the Area Chair**

Dear ACs, SACs, and PCs,

We would like to summarize the status of our rebuttal before the score rollback caused by the reviewer information leak. After the rollback, the visible scores are 8, 4, 4, 2.

However, prior to the rollback:

- Reviewer **TNH4** (initially 2) agreed that our responses addressed their concerns and formally raised their score to **6**.
- Reviewer **aJQS** (initially 4) similarly agreed with our rebuttal and explicitly recommended raising their score to **6**. Although the official edit button had already been disabled at that time, the reviewer clearly stated in the reviewer's comment that “I recommend raising my score to 6.” and also provided detailed reasons for this score increase.
- Reviewer **TQA6** (initially 4) had not yet updated their score but expressed clear positive signals indicating they would likely raise it once their concerns are addressed.

Therefore, the overall scores should reflect **8, 6, 6, 4**, or potentially higher.

Below is a concise summary of each reviewer’s concerns and our corresponding resolutions.

------

### **1. Reviewer TNH4 — Score raised from 2 → 6**

- Real‑world claim → Tasks run on real MCP servers; attacks produce real effects
- Task/domain selection → Justified domain coverage and task representativeness

- Experimental details → Added all settings

- Statistical rigor → Added ANOVA, post-hoc tests, effect sizes, confidence intervals

- Attack Taxonomy → Clarified all categories

**Outcome:** On Nov 22, Reviewer TNH4 explicitly stated: *“I am happy to raise my score.”*
Score updated to **6** before the rollback.

------

### **2. Reviewer aJQS — Recommended raising score from 4 → 6**

- Red-teaming framework connection → Mapped to OTM/CIAP
- Defense discussion →  Added safety‑prompt defense experiment
- Ablations → Added server, decoy tools, attacker budget ablations
- Agent-policy significance → Modified agent strategy

**Outcome:** On Nov 28, Reviewer aJQS concluded: *“I recommend raising my score to 6.”*  and provided specific reasons for the score increase.

------

### **3. Reviewer TQA6 — Score 4, with explicit positive signals**

- Safety-Utility Trade-off → TSR under attack explained quantitatively (r = –0.572, p = 0.041) and qualitatively
- Real-world claim → Tasks run on real MCP servers; attacks produce real effects
- Counter-intuitive SOTA Performance → Reported low TSR reflects performance under attack; clean TSR shows SOTA models perform well.
- Model defense profiles → New flagship models still show spiky vulnerabilities

**Outcome:** Reviewer TQA6 had not yet responded before the rollback, but their official review included clear positive signals that they would consider raising the score if their concerns were resolved. We have full confidence that our rebuttal can resolve all questions.

------

### **4. Reviewer L6g8 — Score remained 8**

Minor clarifications → Provided all requested clarifications promptly.

**Outcome:** Reviewer L6g8 had not yet responded before the rollback, remained positive with score **8**.

------

### **Conclusion**

During the rebuttal, we provided detailed clarifications addressing all concerns raised by the reviewers. We have revised the paper accordingly and updated the PDF on OpenReview. Below is a summary of the revisions.

1. Fixed citation formatting issues and standardized the naming of DeepSeek-V3.1 throughout the paper. (Reviewer TNH4, L6g8, TQA6)
2. Enhanced **Section 2** with an improved positioning of our approach within broader safety frameworks. (Reviewer aJQS)
3. Added brief definitions of attack types in **Table 2**. (Reviewer TQA6)
4. Added additional experimental settings in **Section 4.1**. (Reviewer TNH4)
5. Modified the line colors in **Figure 5**. (Reviewer TNH4)
6. Corrected the trade-off conclusion in **Section 4.2**. (Reviewer TQA6)
7. Replaced hard-to-understand terminology in **Section 4.2** with more accessible expressions. (Reviewer TQA6)
8. Added detailed statistical evidence in **Section 4.2** and **Appendix B** to provide a more thorough and rigorous analysis. (Reviewer TNH4)
9. Analyzed common defensive characteristics across models in **Section 4.2** and **Appendix B.3**. (Reviewer TQA6)
10. Added SAFETY PROMPT defense experiments in **Section 4.3** and **Appendix C**, and discussed additional defense methods in **Section 5.** (Reviewer aJQS)
11. Added additional experiments on the number of servers, decoy tools, attacker budget, and agent strategy in **Appendix D**. (Reviewer aJQS)
12. Added discussions on counter-intuitive SOTA performance, the safety–utility trade-off, and safe refusal in **Appendix E**. (Reviewer TQA6)
13. Updated the data in **Table 4**,**Table 10**.

We respectfully ask the AC to take these revisions and the reviewers’ updated assessments into account in the final decision.

Thank you for your time and effort during this complex review cycle.

---

### Meta-Review · Area_Chair_DD2w · 2026-01-07

**Summary:**

This paper presents MCP-SafetyBench, a benchmark for evaluating safety in real-world Model Context Protocol (MCP) deployments with multi-turn, multi-server workflows. It reveals safety weaknesses in leading LLMs and highlights the need for stronger defenses in agentic systems operating real tools and services.

Initially, the review ratings are mixed, with 8, 4, 4, 2. The authors' rebuttal successfully addresses all of the initial concerns. Before the rollback, the review scores were 8, 6, 6, and 4 (with one reviewer not updating their score). The AC finds the rebuttal convincing. With support from the reviewers, the AC recommends to accept.

**Reviewer Concerns:**

Reviewer aJQS: "Contextualization. Frame the benchmark’s attacker goals/capabilities using a general red-teaming threat-modeling framework"
- The rebuttal provide updated alignment of their work with existing framework. Probably addressed the concern.

Reviewer aJQS: robustness may be agent-policy dependent:
The authors conducted additional experiments comparing two representative agent strategies: Plan-and-Execute (planning-heavy) and ReAct (step-by-step reasoning with tool feedback).

Reviewer TNH4: the detailed analysis of TSR and ASR for each model/domain
The rebuttal provided additional statistical analysis.

**Reviewer Scores:**

Reviewer aJQS: 4: marginally below the acceptance threshold.
-> "Based on these additional proposed changes on contextualization and additional ablations, I recommend raising my score to 6"

Reviewer TNH4: 2: reject, not good enough
-> "Thank you so much for the thorough response! I think that the added statistical analyses make the paper much stronger. I appreciate all the additional data and look forward to seeing them in the next revision. I am happy to raise my score."

Reviewer L6g8:  8: accept, good paper (poster)

Reviewer TQA6: 4: marginally below the acceptance threshold.
-> "I would be happy to raise my rating if these questions are addressed convincingly."

---

### Decision · Program_Chairs · 2026-01-26

Accept (Poster)